# Analytical Methods for the Determination of ^90^Sr and ^239,240^Pu in Environmental Samples

**DOI:** 10.3390/molecules27061912

**Published:** 2022-03-15

**Authors:** Ningjie Zhong, Lili Li, Xiaofan Yang, Yonggang Zhao

**Affiliations:** Department of Radiochemistry, China Institute of Atomic Energy, Beijing 102413, China; zhongnj0314@163.com (N.Z.); lilili@ciae.ac.cn (L.L.); yangxiaofan@ihep.ac.cn (X.Y.)

**Keywords:** environmental sample, ^90^Sr, ^239,240^Pu, chemical separation, analytical methods

## Abstract

Artificial long-lived radionuclides such as ^90^Sr and ^239,240^Pu have been long released into the environment by human nuclear activities, which have a profound impact on the ecological environment. It is of great significance to monitor the concentration of these radionuclides for environmental safety. This paper summarizes and critically discusses the separation and measurement methods for ultra-trace determination of ^90^Sr, ^239^Pu, and ^240^Pu in the environment. After selecting the measurement method, it is necessary to consider the decontamination of the interference from matrix elements and the key elements, and this involves the choice of the separation method. Measurement methods include both radiometric methods and non-radiometric methods. Radiometric methods, including alpha spectroscopy, liquid scintillation spectrometry, etc., are commonly used methods for measuring ^239+240^Pu and ^90^Sr. Mass spectrometry, as the representative of non-radiometric measurement methods, has been regarded as the most promising analytical method due to its high absolute sensitivity, low detection limit, and relatively short sample-analysis time. Through the comparison of various measurement methods, the future development trend of radionuclide measurement is prospected in this review. The fully automatic and rapid analysis method is a highlight. The new mass spectrometer with ultra-high sensitivity shows strong analytical capabilities for extremely low concentrations of ^90^Sr, ^239^Pu, and ^240^Pu, and it is expected to develop determination methods with higher sensitivity and lower detection limit.

## 1. Introduction

With the progress of nuclear energy technology, environmental safety, especially nuclear environmental safety, has become one of the hot issues of common concern for the academic community and the public. Due to human activities, such as nuclear tests, reprocessing of spent nuclear fuel, and nuclear accidents, different artificial radionuclides have been released into the environment. ^90^Sr, as a representative of typical artificial radionuclides with high fission yield and long half-life, is easy to be concentrated in the bones and teeth after ingestion and inhalation by the human body, leading to bone cancer, leukemia, etc. [1]. As an extremely toxic transuranic element with a long half-life of thousands of years, Pu is not only harmful to the environment but also will be enriched in organs after entering the human body [2,3]. As shown in Table 1, the ^239,240^Pu released by the nuclear weapon tests exceeded 10^16^ Bq, and the ^90^Sr was close to 10^18^ Bq, whose radiation pollution increasingly attracts the public’s attention. These radionuclides in the environment can be transferred to the biosphere by dispersion, sedimentation, migration, and accumulation in living organisms through the food chain. It is important for radioactive pollution assessment to monitor the concentration levels of these radionuclides in environmental samples.

The uptake of radionuclides from soil to plants is generally quantified by the ratio of activity concentrations in plant and soil, namely the transfer factor (TF). In addition to radionuclide concentration, one of the factors influencing the TF is the chemical form of the radionuclides, including oxidation state and formation of complexes [5]. The study of the distribution and migration behavior of ^90^Sr and ^239,240^Pu in the environment is important work for radioecology. The migration ability of ^90^Sr is very strong. The TF of Sr from soil to vegetables varied with species, higher than that of other heavy metals. Among them, the value in cabbage was as high as 0.33 [6,7]. The concentration factor (CF) indicates the ability of a biological organism to concentrate radionuclides and is defined as the ratio of the radionuclide content in the organism to the environmental medium. The CF of ^90^Sr by marine organisms was between 0.1 and 10, and it was easy to deposit in shellfish shells, fish scales, and bones [8]. The solubility and mobility of Pu (IV) are relatively low in the pH ranges of 4–10 relevant to the environment ([Pu (IV)_aq_] = 10^−10^ − 10^−8^ mol·L^−1^) [9], while some processes and mechanism, such as the adsorption or complexation of Pu, can increase the mobility. The Pu transport indices for *Brassica juncea* and *Helianthus annuus* increased by addition of diethylenetriaminepentaacetic (DTPA) concentration to prepare Pu-DTPA in the nutrient solution [10]. The solution-to-plant TF for the potato tubers was 0.03–0.80, and the Pu uptake from solution greatly increased by addition of the complexing agent EDTA [11]. Generally, each type of nuclear emission has its own unique isotopic composition, that is, “fingerprint” information (Table 2). Pu in environmental samples is usually derived from global fallout with the value of ^240^Pu/^239^Pu about 0.18. For example, the isotope ratio of ^240^Pu/^239^Pu obtained by Kenna was 0.161684 ± 0.001084 [12]. Understanding the concentration and isotopic composition of ^239^Pu and ^240^Pu in the environment can provide valuable information about nuclear activities in the affected areas. Therefore, accurate measurement of these radionuclides is very important for environmental risk assessment.

The complex matrix of environmental samples with the diverse and unknown existing forms of radionuclides and the trace or ultra-trace concentration level makes accurate measurement quite difficult. Researchers have tried and improved many measurement methods [19,20,21,22,23]. The purpose of this article is to review the separation methods of ^90^Sr, ^239^Pu, and ^240^Pu that have been released in the environment, and discuss commonly used radiometric and non-radiometric measurement methods. By comparing the capabilities of various mass spectrometry technologies, focusing on ICP–MS, the future development of ^90^Sr, ^239^Pu, and ^240^Pu measurements could be deduced.

## 2. Sample Preparation

In radiometric technology of samples, the analysis method can be divided into three steps: sample pretreatment, radiochemical separation (concentration, separation, and purification), and source preparation and measurement. The process is shown in Figure 1. Ultra-trace amounts of the target nuclides ^90^Sr, ^239^Pu, and ^240^Pu in environmental samples need to be separated from a large amount of matrix and interfering nuclides, and this is a great challenge for sample preparation and measurement methods.

### 2.1. Sample Pretreatment

The pretreatment is the most critical step when measuring the contents of ^90^Sr, ^239^Pu, and ^240^Pu in environmental samples. The selection of pretreatment method depends on the purpose of analysis, the method of measurement, and the type of sample. The purpose of pretreatment is to destroy organic matter and convert most of the target nuclides into solution phase, so as to facilitate the subsequent separation operation. Interfering substances should be effectively removed without cross-contamination. Choosing the appropriate pretreatment method and shortening the pretreatment time directly affect the precision and accuracy of the analysis results. At present, the commonly used pretreatment methods mainly contain dry-ashing, digestion, melting, etc.

The traditional dry-ashing method usually uses electric heating plates and muffle furnaces [24,25,26] to dry, carbonize, and ash solid samples (soil, plants, etc.). In order to shorten the ashing time, ashing aids such as HNO_3_, H_2_SO_4_, and H_2_O_2_ can be added. For animal samples that are more difficult to ash and milk samples that contain more water, the microwave-ashing method has more advantages [27]. The powdery sample has a bulky texture and is not easy to ash. The rising of dense smoke during carbonization and the wind of the fume hood will cause partial loss of the components. Adding a small amount of ultra-pure water to the powder sample can effectively improve this phenomenon and enhance the accuracy and reliability of the results [28]. Considering the volatilization loss of nuclides, GB14883-2016 gives recommendations on the maximum ashing temperature of food (Table 3) [29].

The wet-digestion method has been widely used for the analysis of ^90^Sr, ^239^Pu, and ^240^Pu in various environmental samples (Table 4). In the early days, a single acid reagent was often used. Now, mixed acids or acid reagents with H_2_O_2_ are often used to treat samples, such as HNO_3_ + HCl (+ HClO_4_, H_2_O_2_). The conventional wet digestion method has the advantages of rapid oxidation speed, convenient operation, and low loss rate of the nuclides, with a digestion temperature below 300 °C. However, its large amount of acid will increase reagent blanks and is not suitable for processing a large quantity of samples. The microwave-digestion method adopts the pressure principle for digestion, with the advantages of less acid consumption, short time, little loss, low blank value, and easy realization of automatic monitoring. It has been widely used and is suitable for the digestion of biological and geological samples. However, this method has less processing capacity, and it is also not suitable for processing a large number of samples, nor for samples with poor microwave penetration. At present, the microwave digestion–atomic absorption method is widely used to measure trace elements in environmental samples.

In addition, the melting method is sometimes used in the sample pretreatment process [30], such as processing the insoluble residues in the wet-digestion method, accelerating the isotope exchange balance between the tracer and the target nuclide, etc. The melting method can treat kilogram samples, and the treatment effect depends on flux, temperature, and so on.

The same batch of black-tea samples was selected as the research object to investigate the characteristics between different pretreatment methods [31]. The measurement results of different metals by ICP–MS found that the microwave digestion method was significantly better than the dry-ashing method and the wet-digestion method. At present, in order to increase the sample processing capacity, the dry-ashing method is combined with the microwave-digestion method, shortening the processing time and reducing the amount of acid. In order to monitor recovery, a carrier or tracer can be added during digestion.

### 2.2. Chemical Separation

The solution after sample pretreatment contains a large amount of matrix elements, which would interfere with the subsequent accurate measurement. It is necessary to use chemical methods for separation and purification to remove matrix elements and interfering nuclides. There are many chemical separation methods suitable for Sr and Pu, including precipitation, solvent extraction, ion-exchange chromatography, extraction chromatography, or a combination of these methods [36,39,40,41,42,43,44,45,46]. We mainly focus on the ability of various methods to remove elements that interfere with the analysis of ^239^Pu, ^240^Pu, and ^90^Sr.

#### 2.2.1. Separation Methods of ^90^Sr in Environmental Samples

In the sample solution obtained by pretreatment, Sr is usually pre-concentrated by precipitation or co-precipitation via the formation of insoluble nitrates, carbonates, and oxalates. The classical method of strontium precipitation by fuming nitric acid was cumbersome and time-consuming [47]. The precipitate formed by the reaction of alkaline earth metal and carbonate was decomposed with mineral acid in order to separate Sr [48]. Rao et al. [49] used different co-precipitation methods to analyze ^90^Sr in environmental and dietary samples by Ĉherenkov counting of ^90^Y.

Alkaline earth metals, such as Sr, can be well separated from other elements via the ion-exchange method. Dowex-50 is one of the commonly used cation exchangers. Calcium, strontium, and barium were separated by cation-exchange resin, using EDTA with appropriate concentration and acidity as eluent, measuring the ^90^Sr in biological samples successfully [50]. The solvent extraction method is simple and convenient. The extraction rate of strontium by thenoyltrifluoroacetone (TTA) can be as high as 95%, and Bis (2-ethylhexyl) phosphate (HDEHP) can be used to separate ^90^Sr from its daughter ^90^Y. By using dicyclohexyl-18-crown-6 in chloroform solution extraction, a small amount of Sr^2+^ can be separated from a large quantity of Ca^2+^ [51].

A new analytical method was developed for the separation of ^90^Sr from ^90^Y by using DGA resin, in which the extractant system was N,N,N′,N′-tetra-n-octyldiglycolamide (Normal) or N,N,N′,N′-tetrakis-2-ethylhexyldiglycolamide (Branched) [52,53]. The working capacity of the two DGA resins is 7.23 mg Sr and 11 mg yttrium per mL of resin. Sr resin is a special resin for separation and enrichment of Sr, with strong selectivity of Sr, and can efficiently separate Sr from Ca and other elements. Sr resin can replace cation exchange resins for low-consumption (significant reduction in resin consumption and elution amount) and high-efficiency chemical separation, with the extractant of crown ether (4,4′(5′)-di-tbutylcyclohexano-18-crown-6) dissolved in octanol. The adsorption capacity of Sr enhances with the increasing of concentration of nitric acid. The study on the adsorption behavior of elements on Sr resin found that the quantitative recovery of Sr can be basically achieved by loading the column with 3 mol·L^−1^ nitric acid medium and desorbing with 0.05 mol·L^−1^ nitric acid [54]. Željko et al. [32] compared the separation effect of anion-exchange resin, Sr resin, and the combination of the two methods on Sr in environmental samples (soil, vegetation, water, and animal bones) and emphasized their influence on the accuracy of ^90^Sr determination, finding that the efficiency of Sr separation depended on the sample type and separation method.

The interfering nuclide ^90^Zr needs to be decontaminated when using mass spectrometry to measure ^90^Sr. Considering the high concentration of Zr in environmental samples, Sr resin was used for separation, with a 20 *v*/*v*% HNO_3_ system for sample loading and washing to remove 86% of Zr. Meanwhile, the outflow of Sr was about 1% [23]. With deionized water, more than 98% of Sr and about 0.2% of Zr could be eluted from the resin, which could basically separate Sr from the main interfering Zr. Two-stage extraction separation by using Sr resin could further purify Sr, reducing the Zr concentration in solution to below 5 ng·mL^−1^ [55].

#### 2.2.2. Separation Methods of Pu in Environmental Samples

It is necessary to eliminate the interference of ^214^Am, ^210^Po, ^224^Ra, ^229^Th, ^231^Pa, ^232^U, and ^243^Am when using alpha spectroscopy to determine ^238^Pu and ^239+240^Pu. Moreover, the ^238^UH^+^ and ^238^${\mathrm{UH}}_{2}^{2+}$ peaks produced by natural uranium interfere with the measurement of ^239^Pu and ^240^Pu by mass spectrometry. Therefore, after selecting the measurement method, it is advisable to adopt an appropriate separation method to decontaminate the corresponding interfering elements.

Over the past few decades, extraction chromatography has attracted much attention for radiochemical separation and purification of Pu due to its short sample-processing time, small acid reagent dosage, and high selectivity and recovery. Pu in the sample solution presents multiple valence states, namely Pu^3+^, Pu^4+^, ${\mathrm{PuO}}_{2}^{+}$, ${\mathrm{PuO}}_{2}^{2+},$ and ${\mathrm{PuO}}_{5}^{3-}$. Pu (IV) and ${\mathrm{NO}}_{3}^{-}$ or Cl^−^ form stable anionic complexes easily, which is the basis for separation of Pu on anion exchange resins or extraction chromatography. Therefore, it is necessary to adjust Pu to Pu (IV) before loading the column. The most common method is to use oxidants (such as NaNO_2_, ${\mathrm{IO}}_{3}^{-}$, etc.) for one-step value adjustment. It is worth noting that, due to the ultra-trace amount of Pu in environmental samples, Pu (VI) may not be completely reduced to Pu (IV) under conventional experimental conditions, and the unconverted Pu (VI) fraction will be lost during the separation process. Therefore, the most appropriate adjustment method is to reduce Pu to Pu (III) with a reducing agent (such as I^−^ and Fe^2+^), and then use an oxidizing agent to oxidize the generated Pu (III) to Pu (IV) [56].

A variety of extraction resins have been developed at home and abroad, such as TEVA, UTEVA, etc., with large loading capacity and good mass transfer performance, and they are widely used in the sample separation process. TEVA resin [57] is a quaternary ammonium salt, and the stationary phase is trialkylmethylammonium chloride, with a structural formula of CH_3_N(C_n_H_2n+1_)_3_Cl (n = 8–10), generally used for the separation of Np and Pu, with the best adsorption system of 2~3 mol·L^−1^ nitric acid [58]. Using TEVA resin to pre-concentrate Pu in a vacuum box, the decontamination factor (DF) for uranium could reach over 1 × 10^4^, and the DF for thorium and americium was more than 1 × 10^3^ [59]. The Pu solution separated from TEVA resin was further purified by DGA resin with the DF for uranium of 1 × 10^6^, eliminating the interference of the uranium hydrogen peak by measuring ^239^Pu with ICP–MS, and the short-life isotope ^238^Pu was successfully measured by alpha energy spectroscopy [60].

Anion-exchange resins have become attractive methods due to the low purchase cost, and they are widely applied to the separation of Pu for the strong tolerance to matrix elements compared with extraction resins [61]. The commonly used anion-exchange resins are AG-1 × 8, Dowex-1 × 8, etc., with the best adsorption system of 8 mol·L^−1^ nitric acid. Yamato et al. [39] established a combined separation method of Pu and Am in biological samples. The Pu was adjusted to Pu (IV) to make it adsorb on the anion-exchange column, and Am flowed out with the effluent. Hayashi et al. [62] improved it and established an analysis method for ^239/240^Pu in biological samples with a recovery rate of 81.6–91.8%.

The extraction behavior of amine extractants is very similar to that of anion exchangers, with the advantages of rapid extraction speed, high sorption capacity, and high selectivity to Pu (IV) in the nitric acid system. The exchange reaction between tri-n-octylamine (TOA) resin and Pu (IV) is very rapid, and TOA extraction chromatography is a standard method for the determination of ^239^Pu and ^240^Pu in food by alpha spectroscopy [63]. Ji et al. [64] established a process for the separation of Pu by TOA resin and compared it with the anion-exchange resin (Dowex-1 × 8), which had a slightly poorer decontamination effect.

## 3. Measurement Methods of ^239^Pu, ^240^Pu, and ^90^Sr

Radionuclide measurement methods are divided into radioactive measurement methods (such as alpha spectrometry) and non-radioactive measurement methods (such as mass spectrometry). The former is to perform qualitative and quantitative analysis of the target nuclide by measuring the characteristic rays (α, β, or Ƴ rays) emitted by the decay process of radionuclides. Mass spectrometry is the main non-radioactive measurement method, with improved technology, high sensitivity, good precision, and higher equipment popularity than radioactive measurement methods. The various radioactivity measurement methods of ^239^Pu, ^240^Pu, and ^90^Sr are reviewed in this part and compared with mass spectrometry, focusing on mass-spectrometry measurement methods.

### 3.1. Radioactivity Measurement Methods

Different instruments are used to measure different kinds of rays [65]. When a certain nuclide emits different kinds of rays at the same time, the radioactivity obtained by measuring different rays is consistent. The main radiometric methods of nuclides in the environment are shown in Table 5. The radioactive analysis method is the most widely used analysis method for the determination of ^90^Sr, ^239^Pu, and ^240^Pu [21,45]. ^90^Sr is a beta emitter, while ^238^Pu, ^239^Pu, and ^240^Pu are alpha emitters (Table 6).

Common analytical methods for the determination of ^90^Sr in environmental samples are liquid scintillation counting (LSC) and proportional counter [44,67]. Solatie et al. [68] used low-background LSC to measure the concentration of ^90^Sr in soil samples near nuclear facilities, with the activity range of 6.2–96.5 Bq·kg^−1^. Asgharizadeh et al. [69] measured the activity of ^90^Sr in soil and sediment samples from the southern coast of Iran, with the results in the range of 0.40–3.01 Bq·kg^−1^. At the 95% confidence level, the minimum detectable activity (MDA) of the LSC method within a 5-h counting period was 0.33 Bq·kg^−1^. Lee et al. [69,70] measured the activity concentration of ^90^Sr in the volcanic soil of Jeju Island (19.9 ± 2.1 Bq·kg^−1^) and determined that it was higher than that of the Korean Peninsula (8.3 ± 1.7 Bq·kg^−1^) by using low-level LSC. Maxwell [70] reported that the concentration of ^90^Sr in the soil samples from the Fukushima Nuclear Power Plant was 1.35 mBq·g^−1^ by a gas proportional counter. Jabbar et al. [71] used LSC to determine that the average concentration of ^90^Sr in soil samples and plant samples from Islamabad was 4.3 ± 0.0005 Bq·kg^−1^ and 1.05 ± 0.91 Bq·kg^−1^, respectively.

There are many methods for measuring Pu in the environment. After chemical separation, Pu is usually measured by using liquid scintillation counting or alpha spectrometry. However, the separation process of these two methods is more complicated and time-consuming. It usually takes about 1 week to complete a measurement period, and sometimes even more than 10 days. At the same time, the alpha-energy values of ^239^Pu (5.16, 5.14 MeV) and ^240^Pu (5.17, 5.12 MeV) are very close, and this is a great challenge to the energy resolution (at least 25 keV) of the current ion-implanted passivated planar silicon detector [72], leading to the fact that only the total activity of ^239+240^Pu can be measured. Beata Varga et al. [73] used alpha spectroscopy to measure the activity of ^239+240^Pu of grass samples in the hilly area of Budapest as 0.024 ± 0.01 Bq·kg^−1^ (fresh samples), with the MDA of 0.02 Bq·kg^−1^. Compared with the soil surface, the transfer factor (TF) absorbed by roots was 0.032 ± 0.014 (Bq·kg^−1^ fresh grass mass)/(Bq·kg^−1^ dry soil mass), and the TF of forage grass in the literature was in the range of 5 × 10^−5^–0.7. Sherrod et al. [30] determined the activity of ^239^Pu in rice samples as 11.8 Bq·kg^−1^. Lee et al. [74] measured Pu and Am isotopes in the volcanic soil of Jeju Island and the Korean Peninsula. The concentrations of ^239^Pu and ^240^Pu were 2.0 ± 0.3 Bq·kg^−1^ (the error is 1σ) and 0.67 ± 0.1 Bq·kg^−1^, respectively. Liu et al. [58] established a method to quickly determine the content of ^239+240^Pu in environmental water samples by using low-background LSC, and the detection limit of 5 L water samples was 6 × 10^−3^ Bq·L^−1^ within a 60-min counting time. Dagmara et al. [75] studied the distribution of Pu in the organs and tissues of seabirds and found that the highest Pu concentration was in the digestive organs and feathers, while the smallest was in the skin and muscles. The Pu content was lower in fish-eating seabirds, while it was much higher in herbivorous seabirds. It was analyzed that the main source of Pu in seabirds was global atmospheric sedimentation.

In conclusion, the commonly used methods for measuring ultra-trace ^90^Sr and ^239,240^Pu in environmental samples are liquid scintillation counting and alpha spectroscopy. The detection limit of these two methods is low, but the measurement time is long, which is not suitable for emergency monitoring. As alpha spectroscopy can only measure the total activity of ^239+240^Pu, the ratio of ^240^Pu/^239^Pu is unable to be given to analyze the source of Pu.

### 3.2. Mass Spectrometry Methods

Mass spectrometry technology is widely used in the determination of radionuclides with long life and large mass numbers, such as ^235^U, ^239^Pu, and ^240^Pu. For short-lived radionuclides, radioactivity-measurement methods are more commonly used. In recent years, newly developed mass spectrometry techniques with high resolution, high sensitivity, and advanced interference cancellation functions have also been used to measure radionuclides with a shorter life, such as ^90^Sr. This section reviews the mass spectrometry methods, focusing on ICP–MS, to determine ^90^Sr, ^239^Pu, and ^240^Pu in order to better understand the analytical capabilities and potential applications of mass spectrometry.

#### 3.2.1. Measurement Method of ^90^Sr

The natural Sr concentration in environmental samples is about nine orders of magnitude higher than that of ^90^Sr. ^90^Sr is interfered by isobaric ^90^Zr interference and the stable peak trailing interference of ^88^Sr (82.58% abundance) when determined by mass spectrometry. The separation of ^90^Sr and ^90^Zr requires a mass resolution greater than 30,000, which exceeds the resolution capability of mass spectrometry. Researchers have tried many methods. Arslan et al. [76] made the first attempt to determine ^90^Sr by using accelerator mass spectrometry (AMS) after removing ^90^Zr by a chemical procedure. Betti et al. [77] measured the concentration of ^90^Sr in soil, sediment, and grass by glow-discharge mass spectrometry (GD-MS). Wendt et al. [78] reported the concentrations of ^89^Sr and ^90^Sr in soil, grass, milk, and urine after chemical separation by resonance ionization mass spectrometry (RIMS). Bushaw et al. [79] used RIMS to determine ^90^Sr in environmental samples with a detection limit as low as 0.8 fg. AMS, especially RIMS, has high element selectivity. However, they are not usually used in the environmental application, due to less accessibility, and complicated operation and maintenance. ICP–MS, with high sensitivity, can perform rapid and relatively cost-effective isotope analysis at ultra-trace levels [80]. High-resolution inductively coupled plasma–mass spectrometry (HR-ICP–MS) has ultra-high sensitivity and strong interference elimination ability. The ionization energy of Sr (5.70 eV) is 1.14 eV lower than that of Zr (6.84 eV). Operating under cold plasma can greatly reduce the ionization efficiency of Zr by HR-ICP–MS [33]. The isotope ratio of ^90^Sr/^86^Sr in soil samples near the nuclear facility was 6.02 × 10^−9^, basically reaching the sensitivity of radiochemical method for the determination of ^90^Sr [20].

Quadrupole inductively coupled plasma–mass spectrometry (ICP-QMS) [1,23,56] and sector-magnetic-field inductively coupled plasma–mass spectrometry (SF-ICP–MS) [19] are also used for the measurement of ^90^Sr. The interference signal is suppressed by improving the sample introduction method and optimizing the operating system conditions, effectively reducing the detection limit. The concentration of ^90^Sr was measured by tunable bandpass dynamic reaction cell–ICP–MS (DRC–ICP–MS) with relatively high abundance sensitivity [55]. O_2_ was used as the reaction gas to form ZrO^+^ with Zr in the reaction cell, effectively suppressing the interference of Zr. The concentration of ^90^Sr in sediments, plants and water samples was consistent with the activity value determined by Ĉherenkov count and the reference value. The detection limits of the three samples were 0.1 pg·g^−1^ (0.5 Bq·g^−1^), 0.04 pg·g^−1^ (0.5 Bq·g^−1^), and 3 pg·L^−1^ (5 Bq·L^−1^), respectively. Feuerstein et al. [1] used ICP–DRC–MS to quickly determine the concentration of ^90^Sr in contaminated soil samples near the Chernobyl nuclear power plant and compared with the results obtained by the previous radiometry (Table 7). The measurement results were basically consistent, while the accuracy of ICP–DCR–MS was better. Although the MDA of ICP–DRC–MS was inferior to the radiometric methods, it represented a time-saving and economical alternative technology for rapid monitoring of high levels of ^90^Sr contamination in environmental samples.

The interference signals can also be suppressed effectively by using dual mass spectrometry for mass discrimination. ^90^Sr in the reference materials was measured by triple quadrupole collision/reaction cell–CP–MS/MS in single mode and MS/MS mode with relative accuracy and precision of 0.7–3.6% and 3.5–8.1% [34]. The established method was used to measure the artificial radionuclide ^90^Sr in soil samples collected from the whole state of Qatar. The average concentration was 0.606 fg·g^−1^ (3.364 Bq·kg^−1^), with the detection limit of less than 0.04 pg.

#### 3.2.2. Mass Spectrometry Measurement Methods of ^239^Pu and ^240^Pu

As mentioned above, the use of mass spectrometry to measure ^239^Pu and ^240^Pu is interfered by the uranium–hydrogen peak. It is necessary to separate ^238^U before measurement and absolutely eliminate polyatomic chlorides and oxides. Methods such as improving the sample introduction method, increasing the temperature of the plasma, and changing the flow rate of the atomizer can reduce the formation of hydride ions [83]. Among them, a sample introduction method, such as membrane desolvation, is a very important factor that affects hydride [84]. For hydride generation, Kim et al. [85] compared five injection methods in terms of sensitivity, specific sensitivity, and precision. A high-efficiency sample-introduction system Apex-Q combined with the sector field ICP–MS could achieve high instrumental sensitivity and reduce the formation of ^238^UH^+^/^238^U [86].

Thermal ionization mass spectrometry (TIMS) is one of the main methods to determine isotope composition and ratio. The accuracy of isotope measurement is less than 5 × 10^−^^4^, and the abundance sensitivity is better than 5 × 10^−8^. McCarthy et al. [87] measured the isotopic ratios of ^240/239^Pu and ^241/239^Pu in Oxfordshire soil to be 0.184 (RSD < 1%) and 0.0049 (RSD < 5%). The average concentration of ^239+240^Pu was 0.257 Bq·kg^−1^, close to the average value of radioactive deposition in Britain [88]. Buesseler et al. [89] found that TIMS was one order of magnitude more sensitive than the traditional alpha-counting technique when used to detect Pu in sediments, seawater, and coral samples.

TIMS has high precision and abundance sensitivity; however, with a complicated sample-preparation process (requiring pure samples), long measurement time, and a large amount of samples, it is not as simple as ICP–MS to operate. By using the sector-field ICP–MS to measure ^239^Pu, Ni et al. [85] investigated the transfer of global fallout Pu from paddy soil to rice grain in 12 Japanese prefectures, with the TF ranging from 4.5 × 10^−6^ to 1.2 × 10^−4^. Chiappini et al. [90] used high-sensitivity ICP–MS to measure the contribution of Pu deposited in the Mururoa and Fangataufa sediments in the French atmospheric nuclear test for the first time. The contribution was very small and could only be observed in the territorial sea boundary area 22 km away from the atoll. The concentration of Pu in the seawater was only 2 mBq·m^−3^ higher than that observed in Rangiroa. For standard solutions without interference, the detection limits of ICP–MS/HP/Mistral (high-performance ICP–MS associated with Mistral desolvation atomizer (HP/Mistral)) were compared with alpha spectrometry (Table 8). The measurement period of alpha spectrometry was 4000 min. It should be noted that alpha spectrometry measures the sum of the activities of ^239^Pu and ^240^Pu, while the ICP–MS measures the activities of ^239^Pu and ^240^Pu, respectively. On this basis, ICP–MS/HP and alpha spectroscopy were compared for the measured values of ^239+240^Pu in Sellafield sediments, Mururoa lagoon sediments, and grouper samples. These comparisons confirmed that the detection limit of ICP–MS/HP/Mistral was lower than that of alpha spectroscopy when measuring environmental samples. In addition, the introduction of collision/reaction cell (CRC) technology in ICP–MS can effectively reduce the interfering effects of the uranium-hydrogen peaks on measuring ^239^Pu and ^240^Pu. Recently, a CRC–ICP–MS/MS method reduced ^238^${\mathrm{UHO}}_{2}^{+}$/^238^U^16^$O_{2}^{+}$ ratio to 4.82 × 10^−9^, 2-to-3 orders of magnitude lower than previously reported values, and accurately measured the ultra-trace level ^239^Pu with the concentration ratio as low as 10^−10^ by ^239^Pu/^238^U [91].

HR-ICP–MS is also mostly used when measuring the concentration of ^239,240^Pu in the environment. Helal et al. [20] measured the isotope ratio of ^240^Pu/^239^Pu of 0.17 in the soil near nuclear facilities. Stefan et al. [92] used the ultrasonic atomization method to measure the Pu isotope activity and ^240^Pu/^239^Pu isotope ratio in environmental samples, with the measurement accuracy (RSD) of the ratio of ^240^Pu/^239^Pu at about 2%. The total plutonium activity (^239+240^Pu) measured was very consistent with the data obtained by the alpha spectroscopy, with the detection limits of ^239^Pu and ^240^Pu to be 5 and 1 fg·mL^−1^, respectively. Wang et al. [93] measured the activity of ^239+240^Pu in the surface sediments of the Bohai Sea and the North Yellow Sea by the isotope dilution method to be 0.001–0.288 and 0.040–0.269 Bq·kg^−1^, respectively. The atomic ratio of ^240^Pu/^239^Pu was 0.173–0.256 and 0.196–0.275, respectively, slightly higher than the global sedimentation value of 0.18.

In addition, Mathew et al. [94] used AMS to distinguish Pu from the FDNPP accident and Nagasaki explosion from global sediments. It was found that the isotope ratio of ^241^Pu/^239^Pu was more sensitive than that of ^240^Pu/^239^Pu when Pu was traced.

Overall, mass spectrometry has been widely used in the measurement of ultra-trace nuclides. In recent years, CRC–ICP–MS/MS is a promising method with ultra-high sensitivity, which can significantly eliminate the ^90^Zr and ^238^U interferences in the measurement of ^90^Sr and ^239,240^Pu by using O_2_ as the reaction gas.

## 4. Conclusions

Concerns have increased regarding the environmental radioactive contamination occurring in large-scale nuclear or radiological accident/incident. Carrying out relevant monitoring is of great significance, because it reflects the affected degree of biological chain to a certain extent. In this work, we reviewed and summarized the pretreatment, separation, and measurement methods for analyzing ^90^Sr and ^239,240^Pu in environmental samples with the characteristics of ultra-trace concentration and complex matrix. The pretreatment of samples usually costs most of the experimental time. Combining the dry-ashing and microwave-digestion methods greatly increases the processing capacity and shortens the processing time. The classical separation method of Sr is fuming nitric acid precipitation, which is time-consuming and cumbersome. In contrast, the solvent-extraction method is quicker and simpler. In recent years, the commonly used Sr high-efficiency specific resin extraction method has been more selective. As for the separation and purification of Pu in the environment, adjusting the valence state of Pu to Pu (IV) is the primary task. Among the commonly used methods for the separation of trace amounts of Pu, the ion-exchange method is fast and economical, while the exchange capacity of the leaching resin is high. Then we analyzed and compared the nuclear measurement techniques of ^90^Sr and ^239,240^Pu in the environment. The highly sensitive liquid scintillation spectrometry and alpha spectrometry are the general methods for measuring ^90^Sr and ^239,240^Pu, as their instrument maintenance is simpler than mass spectrometry. By improving sample introduction methods and optimizing the operating conditions of the instrument, mass spectrometry has the ability to successfully measure the concentrations of ^90^Sr and ^239,240^Pu in the environment with equal sensitivity and a lower detection limit to radiometric methods.

The pretreatment of a large number of samples and the low detection limit of the instrument are required for the complex matrix of environmental samples and the ultra-trace level of radionuclides. Here are some prospects for the radionuclide analysis of environmental samples. First, as sample preprocessing is often cumbersome and time-consuming, it is critical to establish a rapid, convenient, efficient, and clean pretreatment method in the future. Second, based on the characteristics of radionuclides in environmental samples, research on developing high-capacity, high-efficiency, and specific separation resins for major beta nuclides and actinides is the development trend. Providing rapid and ultra-sensitive analysis technology is of great significance to environmental emergency monitoring. Therefore, automatic separation and measurement technology will become a promising direction due to its fast and convenient advantages. In addition, research on mobility of ^90^Sr and ^239,240^Pu in the environment is the basis for predicting the degree of ecological environment pollution, as well as the restoration study of soil, vegetation, and water after contamination with these nuclides.

## Figures and Tables

**Figure 1 molecules-27-01912-f001:**
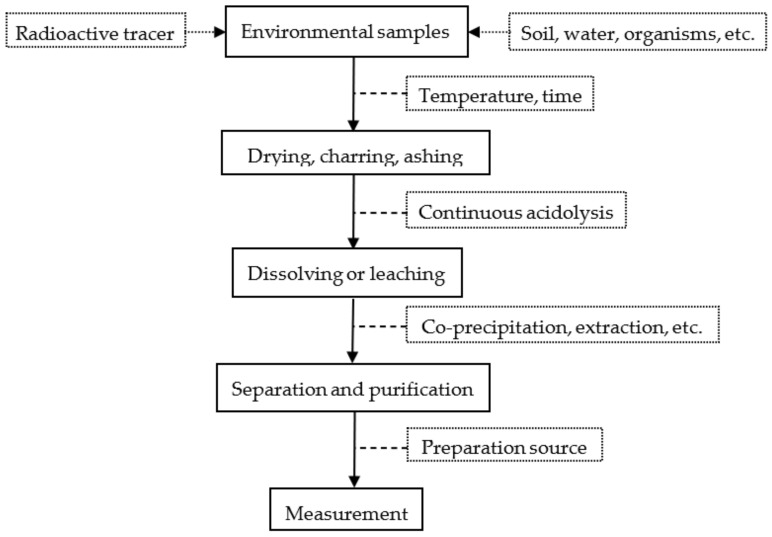
Separation process of radionuclides in environmental samples.

**Table 1 molecules-27-01912-t001:** Source and total release of artificial radionuclides (PBq). Adapted with permission from ref. [4].

Radionuclide	Weapon Tests		Chernobyl Accident		Fukushima Accident	
	Atmosphere	Ocean	Atmosphere	Ocean	Atmosphere	Ocean
^131^I	-	-	1760	-	150–160	-
^137^Cs	950	600	85	16	12–20	4–27
^90^Sr	600	380	1	-	0.01–0.14	0.1–2.2
^239,240^Pu	10.87	6.6	0.087	-	(1–2.4) × 10^−6^	-

**Table 2 molecules-27-01912-t002:** Pu isotope ratios in pollution from different nuclear events.

^240^Pu/^239^Pu	Ratio	References
Spent fuel reprocessing	0.02–0.06	[13,14]
Weapons-grade	<0.06	[15]
Global fallout	0.176 ± 0.014	[16]
Fukushima	0.330–0.415	[17]
Chernobyl	0.4	[18]

**Table 3 molecules-27-01912-t003:** Maximum ashing temperature of food samples specified in GB14883-2016.

Analysis Project	^89^Sr, ^90^Sr	^137^Cs	^147^Pm	^226^Ra	Natural Thorium	Natural Uranium	^239^Pu
**Temperature (°C)**	550	450	450	550	550	550	450

**Table 4 molecules-27-01912-t004:** Digestion methods applied to the analysis of ^90^Sr, ^239^Pu, and ^240^Pu in the environment.

Research Object	Acid Reagent	Digestion Equipment	References
10 or 20 g soil sample	1 mL Concentrated HNO_3_, 150 mL Deionized water		[32]
5 g plant sample ash	100 mL 5 mol·L^−1^ HNO_3_	Infrared Lamp	[32]
1 g soil sample	4.8 mL HF, 1.2 mL HClO_4_	Microwave Digestion Apparatus	[33]
50 g reference material + 500 g Qatari soil	500 mL Concentrated HNO_3_, 250 mL HCl	Electric heating plate	[34]
0.5 g sludge ash sample from sewage treatment plant	12 mL HNO_3_:HCl (3:1) mixture	Microwave Digestion Apparatus	[35]
5 g dried food	1:1 HNO_3_, H_2_O_2_	Electric heating plate	[36]
0.5 g food sample	Concentrated HNO_3_	Microwave Digestion Apparatus	[37]
1 g soil sample	4.8 mL HF, 1.2 mL HClO_4_	Microwave Digestion Apparatus	[38]

**Table 5 molecules-27-01912-t005:** Main radioactive detection methods of radionuclides in environmental samples.

Analytical Method	Ray Type	Main Detection Nuclides	Advantage	Disadvantage
Alpha spectrometry	α	^238^Pu, ^239^Pu, ^210^Po, ^241^Am	Low detection limit and high sensitivity	Complicated process and time-consuming
Beta spectrometry	β	^3^H, ^89^Sr, ^90^Sr, ^226^Ra, ^137^Cs	Low background and high detection efficiency	Tedious preprocessing and significant spectrum interference
Gamma spectrometry	γ	^55^Fe, ^60^Co, ^65^Zn, ^95^Zr, ^110^Ag, ^131^I, ^134^Cs, ^137^Cs	Simple preprocessing, high detection efficiency, and strong energy resolution	High cost and many influencing factors
Liquid scintillation	α, low energy β	^3^H, ^14^C, ^89^Sr, ^90^Sr, ^239^Pu	High detection efficiency, high sensitivity, and high precision	Complicated separation and time-consuming

**Table 6 molecules-27-01912-t006:** Half-life, decay mode, and strongest decay energy of ^90^Sr, ^238^Pu, ^239^Pu, and ^240^Pu [66].

Nuclides	Half-Life	Main Alpha Particles		Main Gamma Rays		Main Beta Particles	
		Energy/keV	Intensity/%	Energy/keV	Intensity/%	Energy/keV	Intensity/%
^90^Sr	28.9 ^a^*					545.9	100.0
^238^Pu	87.7 ^a^	5499.03	70.91	43.498	0.0392		
^239^Pu	24,110 ^a^	5156.59	70.77	51.624	0.0272		
^240^Pu	6561 ^a^	5168.17	72.8	45.244	0.0447		

* ‘a’ represents the unit of years.

**Table 7 molecules-27-01912-t007:** Determination of ^90^Sr activity in soil samples by ICP–DRC–MS and radiometric method [81,82].

Sample	ICP–DCR–MS		Radiometric Method	
	(pg·g^−1^)	(Bq·g^−1^)	(Bq·g^−1^) 1996	(Bq·g^−1^) 2007
soil 1	4.66 ± 0.27	23.7 ± 1.3	45 ± 9	35 ± 7
soil 2	13.48 ± 0.68	68.6 ± 3.5	82 ± 16	63 ± 13
soil 3	12.9 ± 1.5	65.6 ± 7.8	99 ± 18	76 ± 15

**Table 8 molecules-27-01912-t008:** Instrument detection limits of ICP–MS/HP/Mistral and alpha spectroscopy [90].

Isotope	ICP–MS/HP/Mistral		Alpha Spectrometry
	Mass/g	Activity/Bq	Activity/Bq
^239^Pu	1.2 × 10^−15^	2.8 × 10^−6^	1 × 10^−4^
^240^Pu	1.2 × 10^−15^	1.0 × 10^−5^	1 × 10^−4^

## Data Availability

Not applicable.

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
