# Peer review of "Analytical Methods for the Determination of 90Sr and 239,240Pu in Environmental Samples"

_molecules, 2022, doi:10.3390/molecules27061912_

Round 1

Reviewer 1 Report

Altough the paper does not bring new information, it is a good short review if one wants to start to deal with 90Sr and Pu in environmental samples of several kinds. Number of references is adequate (90+) and include both research verified by years of applications and new trends and techniques. I recommend it for publishing and have only minor suggestions / corrections:

L16 ...239+240Pu and 90Sr...
L32 Please specify the value of the 90Sr half-life.
L34 Please specify the value of the half-life for the discussed Pu isotopes.
L49 ...90Sr and 239,240Pu...
L95 "etc." instead of "et al."
L170 ...chemical...
L250-251 I would use the terms "beta emitter" and "alpha emitter" instead of beta and alpha radionuclide.
L264 ...4.3±0.0005 Bq·kg-1 and 1.05±0.1 Bq·kg-1...
L310-315 Add the reference number after the author name, e.g., Arslan [76]. Add "et al." if the referenced literature was co-authored by more people, e.g. Arslan et al. [76].
L315-317 Please re-write the sentence so its easier to understand.
L334-335 The detection limits of the three samples were 0.1 pg·g-1 (0.5 Bq·g-1), 0.04 pg·g-1 (0.5 Bq·g-1) and 3 pg·L-1 (5 Bq·L-1), respectively.
L343 Add reference(s) to Table 7.
L347-351 This paragraph is about 90Sr, the Pu isotopes should not mentioned here; its confusing.
L364-365 ...less than 0.0005... (or 5x10-4)
L374 Ni et al. [83]...
L385 ...240Pu, respectively.
L390 What is CRC in CRC-ICP-MS? Please specify.
L394 Add reference(s) to Table 8.
L395 Add reference(s) to Table 9.
L396 ...used when...
L397 Helal et al. [20]...
L378 Stefan et al. [90]...
L402 ...and 1 fg·mL-1, respectively.
L404 ...and 0.040–0.269 Bq·kg-1, respectively.
L405 ...0.196–0.275, respectively.
L459 Re-check the reference numbers in the list and text because there are some differences, e.g., Stefan et al. is [90] in the list while being [89] in the text.
L607 I recommend finding a new ref. for [72]. Accessing an url more than 20 years ago does not look good...

Reviewer 2 Report

In the review article "analytical methods of 90Sr, 239,240Pu in environmental samples", the authors have given a comprehensive report about the ongoing challenges in detecting and quantifying long-lived radionuclides in the environment due to global accidents/incidents, and the various methods currently used to prepare, pre-treat such samples to separate and quantify the radionuclides of interest. I think that the review is well structured and informative with relevant tables and figures added when appropriate. 

However, there is a lot of scope of improvement in the language used and several sentences are poorly constructed to the point of being misleading. 

Here are the changes I suggest before accepting;

Title: I would recommend changing the title to " Analytical methods for the determination or quantification (whichever deemed appropriate) of ......samples"

Line 7: Change to "Artificial long-lived radionuclides such as.....have long been released.."

Line 14: Change to "include both radiometric and non-radiometric methods"

Line 22: Change to "develop quantification methods with higher sensitivity..."

Line 30: Remove "quantities of"

Line 33: Change to "after ingestion and inhalation by the human body leading to bone cancer"

Table 1: Insert space between "Weapontests"

Line 52: Kindly define "concentration coefficient"

Line 57-58: Clarification needed. Did the transport index increase by adding the Pu-complex or just the complex?

Line 70-71: Change to "concentration level makes accurate measurement quite difficult"

Line 73: Change to "released in the environment.."

Line 74: Remove "with"

Line 81: Change to "Ultra trace amounts of the target nuclides 90Sr...in environmental samples need to be separated from ...."

Figure 1: Change the caption to "The separation process of radionuclides in environmental samples"

Line 91: Change "system" to "phase"

Line 148: Replace "such as" with "via"

Line 199: Change to "Pu (IV) and ....... form stable complexes easily,..."

Line 212: Should the "et" in the formula be "Et" as in the ethyl group?

Line 227: Replace "of analyzing" with "for".

Line 232: "alpha" or "α", be consistent throughout the paper.

Line 241: Replace "mature" with "improved"

Line 251: Make radionuclides, plural

Line 282 and elsewhere as appropriate: Add "respectively"

Line 310: Please define "AMS" here.

Line 317: Change to "..due to less accessibility, and complicated operation and maintenance"

Line 375: Replace "ranged" with "ranging"

Line 386: Should be "compared for..."

Line 417: Change to "contamination occurring in large scale nuclear or..."

Line 419: Replace "summary" with "summarize"

Line 452: Clarification needed. "restoration of soil, vegetation and water pollution"????
